# Study on Morphology, Microstructure and Properties of 6063-T6 Aluminum Alloy Joints in MIG Welding

**DOI:** 10.3390/ma16134886

**Published:** 2023-07-07

**Authors:** Shuwan Cui, Yunhe Yu, Rong Ma, Fuyuan Tian, Shuwen Pang

**Affiliations:** 1School of Mechanical and Automotive Engineering, Guangxi University of Science and Technology, Liuzhou 545006, China; yuyunhe1997@163.com (Y.Y.); tianfuyuan2021@163.com (F.T.); pangshuwen273437@163.com (S.P.); 2Dongfeng Liuzhou Automobile Co., Ltd., Liuzhou 545005, China; 3Guangxi Liugong Machinery Co., Ltd., Liuzhou 545006, China; mar@liugong.com

**Keywords:** MIG welding, numerical simulation, 6063-T6 aluminum alloy, microstructure, joint properties

## Abstract

In this paper, a metal inert gas (MIG) shielded welding method was used for high-quality welding of 6063-T6 aluminum alloy sheet with a thickness of 2.5 mm. The welding process of MIG welding was accurately simulated and the welding temperature field and thermal cycle curve were calculated using a combination of Gaussian body heat source and double ellipsoidal heat source. As the welding current increased from 75 A to 90 A, the reinforcing phase precipitated under the microstructure of the joint gradually became larger and re-solidified into the body, resulting in a reduction in mechanical properties. When the welding current is 85 A, the pitting resistance of weld forming and weld area reaches its optimum. At this time, the tensile strength of the joint is up to 110.9 MPa, the elongation is up to 16.3% and the Vickers Microhardness is up to 46.9 HV.

## 1. Introduction

The 6XXX series aluminum alloy is an aluminum alloy with magnesium and silicon as the main alloying elements. It has the advantages of high strength, reliability in processing, corrosion resistance and good heat treatment properties, and is widely used in various fields such as aerospace, automotive engineering, construction, marine engineering, electronics and transportation [1,2,3,4,5]. Aluminum alloys are lightweight and inexpensive, and are widely used in the manufacture of structural parts for automobiles to achieve light weight [6]. MIG welding is a welding method that uses inert gas to protect the weld metal from contamination. It has the advantages of fast welding speed, high welding quality, simple operation, high productivity and high efficiency. TIG (Tungsten Inert Gas) welding also has unique applications in aluminum alloy welding, but MIG welding is more advantageous in high-speed welding of large materials, and the quality of the welds obtained is better [7,8]. Under the effect of the welding thermal cycle, the melted hardening phases in 6063-T6 aluminum alloy precipitate again. The microstructure of the welded joint changes, which affects the mechanical properties and corrosion resistance of the welded joint [9,10,11]. Therefore, the study of microstructure evolution mechanism of 6063-T6 aluminum alloy welded joint is the key to further improve the properties of the welded joint.

To date, many scholars have carried out research on the microstructure and properties of 6063-T6 aluminum alloy welded joints. Sakthivel et al. [12] studied the MIG welding behavior of 6063, 7075 and other types of aluminum alloys with low carbon steel, and the mechanical properties tests and microstructure observation and other experimental tests showed that the weldability and strength of 6063 aluminum alloy was better. Fan et al. [13] used a laser MIG composite welding process to weld 3 mm thick 6061 aluminum alloy plates. Reliable welding parameters were determined experimentally and the fatigue fracture characteristics of welded joints at different stress amplitudes were investigated, and porosity was found to be the main factor leading to joint fracture. Although the microstructure and mechanical properties of welded joints have been extensively studied, the mechanism of microstructure evolution on the joint properties has not been described. Therefore, we have to further analyze the mechanism of microstructure evolution and explore its influence on the performances of welded joints. The microstructure of the welded joint is mainly affected by the welding thermal cycle, and in the actual welding process it is difficult to obtain the actual temperature distribution information by experimental means, so the method of numerical simulation of the welding process is very necessary.

Much work has been carried out by scholars in the numerical simulation of welding. Zheng et al. [14] used finite element simulation techniques to model and simulate the welding process and numerically simulate the ultrasonic impact treatment (UIT) process. The results showed that the numerical simulation technique can better predict the residual stress distribution in the weld and the UIT technique can improve the residual stress in the weld. Although the above-mentioned scholars have successfully predicted the distribution of residual stress through numerical simulation techniques, the physical and chemical phenomena involved in the welding process are very complex and are affected by a variety of factors. The mechanical properties of the welded joint degradation are mainly due to the microstructural changes caused by the action of the welding thermal cycle, and thus in order to obtain the mechanism of the impact of the welding thermal cycle on the welded joint, it is necessary to understand the distribution of the temperature field of the welded joint, and it is thus necessary to analyze the numerical simulation of the temperature field of the MIG welding process of 6063-T6 aluminum alloy [15,16,17].

At present, most researchers only pay attention to the effect of microstructure on the mechanical properties of welded joints, and the research on the pitting resistance of joints is rare. When pitting occurs, the microstructure of welded joints will be irreversibly damaged, which will affect the mechanical properties of welded joints [18]. Because of the role of the welding thermal cycle, the welded joint will inevitably present the precipitation phase and elemental content changes; these changes will cause deterioration in the mechanical properties of the welded joint and corrosion resistance [19,20,21]. The 6063-T6 aluminum alloy has good pitting resistance, but different welding methods produce large heat inputs and different Mg and Si composition ratios; these factors will increase the pitting resistance sensitivity of aluminum alloy. Therefore, reasonable adjustment of welding heat input can significantly improve the pitting resistance of welded joints.

At present, there are few studies on the numerical simulation of MIG welding of 6063-T6 aluminum alloy. In this paper, the relationship between the microstructure evolution mechanism of welded joints and mechanical properties and corrosion resistance is systematically studied by combining the results of numerical simulation analysis with experiments.

## 2. Experimental Procedure

### 2.1. Materials

In this paper, a 2.5 mm thick 6063-T6 aluminum alloy plate is used as the base material. Although the chemical composition of ER5356 welding wire is slightly different from that of the base material, it is widely used for MIG welding of aluminum alloys in the actual production process and has excellent reliability [22]. The chemical composition and mechanical properties of the base metal and welding wire are shown in Table 1 and Table 2. JMATPRO 8.0 software was used to calculate the thermal conductivity, specific heat capacity and density, and the trend of their values changing with temperature is shown in Figure 1.

### 2.2. Experimental Equipment

The experiments were carried out using the MIG welding technique, using 1.2 mm diameter wire and longitudinal butt welding of thin aluminum alloy plates with dimensions of 300 mm × 100 mm × 2.5 mm. The welding system consists of a TransPuls Synergic 4000 CMT welder and an ABB robot. The welding schematic is shown in Figure 2, and the welding process parameters are shown in Table 3. EDM (Electrical Discharge Machining) wire cutters were used to create corrosion specimens, and different types of sandpaper were used to grind and polish the specimens before removing the oil with acetone. The processed patterns were then cleaned using anhydrous ethanol and ionized water and dried naturally in air. In accordance with the ASTM E407-3 standard, the polished surface was etched with Keller’s reagent (2 mL HNO_3_ + 3 mL HCL + 5 mL HF + 150 mL H_2_O) for about 18 s. The microstructure of the welded joints was observed using a metallographic microscope (OM Axioscope 5) and a field emission scanning electron microscope (ZEISS Sigma HD VP, Oberkochen, Germany).

### 2.3. Mechanical Property Test

Tensile tests were carried out using a tensile tester (Wance EMT204C) and were based on the ASTM-E8 standard for mechanical tensile testing. The tensile specimens are shown in Figure 3, with a stretching speed of 3 mm/min and three samples tested under each condition. A Vickers Microhardness tester was used (HVS-1000Z) to measure point by point from the center of the weld toward the base material, with a loading load of 0.5 kgf and a dwell time of 10 s at a spacing of 0.5 mm per point.

### 2.4. Electrochemical Experiments

Electrochemical experiments on the WZ of welded joints were performed with an electrochemical workstation (CHI660E). The platinum electrode was selected as the auxiliary electrode and the saturated glyceryl electrode was used as the reference electrode. The intercepted weld zone specimen was used as the working electrode with a working area of 0.06 cm^2^, inlaid in epoxy resin, as shown in Figure 4. The corrosion liquid was 3.5% NaCl solution under deoxidizing conditions; the scanning potential range was −1.4 V~−0.5 V, and the scanning speed was 1 mV/s.

## 3. Numerical Simulation

### 3.1. Basic Theory of Welding Temperature Field

During the welding heat transfer process, the temperature changes sharply, and the thermophysical properties of the material also change, and there are melting and phase transitions at the same time. Researchers have experimentally determined the fundamental law of heat conduction—Fourier’s law [23], expressed as Equation (1):(1)$$q=-\lambda \left(\frac{\partial T}{\partial x}i+\frac{\partial T}{\partial y}j+\frac{\partial T}{\partial z}k\right)$$
where *λ* represents thermal conductivity, *T* represents temperature, *x*, *y* and *z* are coordinates, and *i*, *j* and *k* are the unit vectors on the three axes.

The thermal conductivity, specific heat capacity and density need to be obtained during the numerical simulation of the temperature field. According to the actual situation in the welding process, the Fourier differential equation is chosen to represent the non-linear three-dimensional transient temperature field, taking into account the non-linearity of the welding temperature field simulation [24], expressed as Equation (2):(2)$$c\rho \left(\frac{\partial^{2}T}{\partial x^{2}}+\frac{\partial^{2}T}{\partial y^{2}}+\frac{\partial^{2}T}{\partial z^{2}}\right)+q$$
where *c* represents specific heat capacity, *ρ* refers to density, *k* is the thermal conductivity, *q* represents heat flow density, *T* is temperature.

The heat source is in a constant state of motion during the welding process and the material within and around the base material is constantly and dramatically changing, and thus two main types of heat transfer are considered: heat radiation and heat convection. Newton’s law and the Stefan–Boltzmann law are used to represent the exchange of heat [25,26,27], expressed as Equation (3):(3)$$q=-\sigma \varepsilon \left(T^{4}-T_{a}^{4}\right)-\left(T-T_{a}\right)$$
where *q* denotes heat loss, *𝜀* represents emissivity, *σ* represents the Stefan–Boltzmann constant, $T_{a}$ is the ambient temperature, *T* is the reference temperature.

### 3.2. Heat Source Model

A suitable heat source model will provide a more accurate simulation of the welding process. MIG welding produces a large amount of energy and has a high aspect ratio, with a large heat flow along the thickness of the base material, so a bulk heat source model will be more accurate. The welding process is simulated using both a single heat source model and a combined heat source model, with the heat source loading shown in Figure 5.

#### 3.2.1. Mesh Model

In order to balance the calculation time and calculation accuracy, we built a simplified model consisting of two plates with dimensions of 100 mm × 80 mm × 2.5 mm. The mesh of the weld and its nearby area is dense, and the minimum mesh size is 1 mm × 1 mm × 1 mm, whereas the mesh of the area far from the weld is sparse, and incremental mesh is used to increase its mesh cell size. After dividing the whole model, the total number of nodes is 49,235.

During the actual welding process, the movement of the arc along the welding direction results in an asymmetrical distribution of the arc heat flow. At the same time, due to the influence of welding speed, the heated area in front of the arc is smaller than that behind the arc, resulting in the heated area not being a single semi-ellipsoid with symmetry across the center line of the arc, but a double semi-ellipsoid. The shape of the semi-ellipsoid before and after the arc also differs, as shown in Figure 6 [28,29,30].

The power density functions of the anterior and posterior ellipsoids are represented by Equations (4) and (5), respectively [31]:(4)$$q_{1}\left(x,y,z\right)=\frac{6\sqrt{3}f_{1}Q}{\pi^{\frac{3}{2}}a_{1}bc}\exp\left(-3\left(\frac{x^{2}}{a_{1}^{2}}+\frac{y^{2}}{b^{2}}+\frac{z^{2}}{c^{2}}\right)\right)$$
(5)$$q_{2}\left(x,y,z\right)=\frac{6\sqrt{3}f_{2}Q}{\pi^{\frac{3}{2}}a_{2}bc}\exp\left(-3\left(\frac{x^{2}}{a_{2}^{2}}+\frac{y^{2}}{b^{2}}+\frac{z^{2}}{c^{2}}\right)\right)$$

In the formula, $f_{1}$ and $f_{2}$ are the heat input shares in the front and rear semi-ellipsoids, respectively; $f_{1}\;$*+*$\;f_{2}$ = 2. $a_{1}$, $a_{2}$, *b*, and *c* are the lengths of the front and rear hemispheres, the width of the molten pool, and the depth of the molten pool, respectively, and *Q* represents the total power introduced.

#### 3.2.2. Gaussian Body Heat Source Model

The Gaussian body heat source model has a Gaussian distribution of radial heat flow with a constant heat flow along the thickness direction of the base material and a cylindrical heat source endogenous to the workpiece [32]. The heat source model is shown in Figure 7.

The function of the Gaussian body heat source model is represented by Equation (6) [33]:(6)$$q\left(x,y,z\right)=\frac{3P}{\pi r_{0}^{2}h}\exp\left(\frac{-3\left(x^{2}+y^{2}\right)}{r_{0}^{2}}\right)u\left(z\right)$$

In the formula, *P* is the total power introduced, *r*_0_ is the effective radius of the heat source, *h* is the effective depth of the heat source and $x^{2}+y^{2}$ is the distance from any point on the model to the center of the heat source. *u(z)* is the unit step function, expressed by Equation (7), and *H* is the thickness of the weldment:(7)$${\{}_{u\left(z\right)=1,0\leq z\leq h}^{u\left(z\right)=0,h\leq z\leq H}$$

## 4. Results and Discussion

### 4.1. The Morphologies of MIG Welds

The morphologies of the MIG welds under different welding currents are shown in Figure 8. When the welding current was 75 A, there were porosity defects on the front of the weld, and most of the back was not fully penetrated. This is due to low welding current, resulting in poor arc stiffness and inadequate arc burning. When the welding current was 80 A, the front side of the weld was well-formed, and the back part was unevenly formed. When the welding current was 85 A, the front of the weld was clearly arranged in a fish scale pattern, the fusion was good and there were no porous cracking defects, and the back of the weld was well-formed and uniformly full. When the welding current was 90 A, the front side of the weld completely collapsed and over burning occurred, and the back side of the weld was too large. Based on the above experimental results, it can be seen that when the welding current was 80 A~90 A, the morphology of the welds was better.

### 4.2. Numerical Simulation of 6063-T6 Aluminum Alloy MIG Welding

#### 4.2.1. Verification of Simulation Results

The appearance of the welded joint obtained by numerical simulation of three heat source models is shown in Table 4. According to the comparison between the simulated weld pool section and the actual weld pool section, the lower half of the weld pool becomes narrower under the action of a single double ellipsoid heat source. Under the action of a single Gauss body heat source, the molten pool has a cylindrical shape. Both of these are inconsistent with the actual welding condition. When the double ellipsoidal heat source was combined with the Gauss body heat source, not only was the workpiece completely penetrated in the numerical simulation, but also the curve of the simulated molten pool boundary was consistent with the actual molten pool boundary.

#### 4.2.2. Distribution Characteristics of Temperature Field in the MIG Weld

In order to further understand the characteristics of the temperature change during the welding process, after a welding time greater than 10 s and the welding heat source tended to stabilize, a total of five points were determined along the centerline of the weld and perpendicular to the centerline of the weld at 1 mm intervals, and were named point 1–point 5. The constant movement of the welding heat source causes the temperature at a point on the welded workpiece to change from a low to a high temperature over time, and then from high to low when the maximum value is reached, in the process known as welding thermal cycling. As shown in Figure 9, thermal cycling curves are obtained by selecting five points on the simulated cross-section of the welded joint, which gives more insight into the characteristics of the changing temperature field. The temperature rises sharply at the start of the weld and reaches the melting point of the experimental sheet, and then starts to fall sharply after the temperature has reached its peak, which is influenced by the convective heat transfer from the air.

Under the same welding current, the time required for each point to reach the highest temperature is almost the same: about 6 s. The point near the center of the weld can reach a maximum temperature of 1190 °C. Different points at different locations have different rates of temperature rise and fall due to different cooling environments. Figure 9 indicates that the slope of the temperature curve at each experimental current is large until the peak temperature is reached, and that the slope of the curve decreases after the peak is reached, which indicates that the melting rate of the weld base material is greater than the cooling rate. The cooling rate during the welding process will affect the precipitation of the second phase of 6063-T6 aluminum alloy. If the cooling rate is too slow in the case of higher current, the precipitated phase will re-solidify inside the matrix and the grains will continue to grow, thus leading to a decrease in the mechanical properties of the welded joint.

### 4.3. Mechanical Properties

#### 4.3.1. Micro-Hardness Testing

The microhardness of 6063-T6 aluminum alloy in the welded joints at different welding currents are shown in Figure 10. The microhardness distribution curve of the four groups of welding current has the same trend, and the lowest microhardness of the joint WZ is 25.2 HV, which is related to the typical casting structure of the welded joint WZ. As the welding current increases, the microhardness of the welded joint had a tendency to decrease. The welding current was too low when the plate was not completely melted through; when there was only a small amount of reinforced phase solid solution of the matrix inside, the solute element density was low and led to a decrease in the microhardness of the welded joint. Excessive welding current would lead to a high temperature of the thermal cycle, when Mg, Si and other atoms dissolved into the internal body to form a supersaturated solid solution resulting in a decrease in the microhardness of the welded joint. When the welding current was 85 A, the degree of dissolution of the strengthening phase was very high, and it was completely dissolved inside the matrix to form a saturated solid solution; thus the microhardness of the welded joint was optimum.

#### 4.3.2. Mechanical Properties

The tensile strength of the experimental base material and the tensile specimens at different currents were tested. Figure 11a indicates the tensile curves of the base material and the tensile specimens at different currents, and Figure 11b illustrates the tensile strength and elongation of the base material and the tensile specimens at different currents. Figure 11b shows that the 6063-T6 base material has a tensile strength of 153.9 MPa and an elongation of 14.9%. The experimental results show that when the welding current was between 75 A and 85 A, the tensile strength and elongation both showed an upward trend, but when the welding current was 75 A, the tensile strength, and elongation did not meet the requirements; when the welding current was 85 A, the tensile strength reached 110.9 MPa and the elongation reached the highest value of 16.3%, both of which are the highest level and meet the mechanical property standards. When the welding current was 90 A, the tensile strength of the sample decreased to 96.5 Mpa and the elongation decreased to 12.4%, the specific experimental results are shown in Table 5.

The tensile fracture of the material specimens mainly consists of fine equiaxed tough nests, which are small and deep in size. The tough nests are of a uniform size and show ductile fracture characteristics. The center of the fracture is mainly distributed by fine isometric tough nests, in which the tough nests are small. Inclusions or second-phase particles in the matrix can be found in some tough nests, as shown in Figure 12. It was found that the tensile strength and microhardness of the specimens were higher when the tensile fracture was a fine equiaxed tough nest, and the phenomenon was mainly related to the hardening effect of the precipitated phase.

When the welding current was low, the tensile fracture was distributed with several pores of different sizes and a small amount of plastic slippage surface, as shown in Figure 13a. The presence of pores leads to a reduction in the effective load-bearing area of the welded joint, which to some extent weakens the strength of the joint and makes it a weak region. Plastic deformation occurs under the action of tensile stress, and at the stress concentration or porosity defects, the microscopic void nuclei grow and gather, forming a tough nest when fractured. In such cases, the tough nest size was not uniform and there were traces of tearing, so the strength and toughness were low.

When the welding current was 80 A, the fracture surface of the tensile specimen was mainly dominated by tough nests, showing ductile fracture characteristics. Compared with the parent material, the size of the tough nests was significantly larger and shallower, their distribution per unit area was relatively reduced, and they had a less pronounced shape, as shown in Figure 13b. When the welding current reached 85 A, the tensile fracture was dominated by equiaxed tough nests, and had relatively small size, depth and tensile strength, and elongation was also improved, as shown in Figure 13c. When the welding current was 90 A, the tensile fracture tough nest size was coarser and varied in size; the number of tough nests per unit area was reduced; the tensile fracture appeared as local tearing phenomenon; and in the macroscopic view was manifested in the reduction in tensile strength and elongation, as shown in Figure 13d.

### 4.4. Microstructure

The 6063-T6 alloy is an Al-Mg-Si series aluminum alloy, which is a heat-treatable strengthened aluminum alloy. The main strengthening phase within the matrix is the β-phase, and the grains of the base material are fibrous. The strengthening phase Mg_2_Si (black particles) is distributed in the base material matrix, as shown in Figure 14.

Although the number of different welding currents under the grain boundary eutectic organization does not vary significantly, a large difference exists in its morphology, as shown in Figure 15. When the welding current was large, the grain boundary eutectic structure was distributed in a coarse grid; when the welding current was small, the eutectic structure had a discontinuous rod-like distribution, and a large amount of eutectic structure was dispersed in the grain. The phase composition of 6063-T6 aluminum alloy weld metal is mainly α-Al solid solution. When the welding pool cools down, the liquid-solid phase transformation begins, forming a small amount of α-Al solid solution. As the temperature decreases, the volume fraction of α-Al solid solution increases.

During the welding process, the cooling rate of 6063-T6 aluminum alloy is faster, which will inhibit the second phase precipitation. When the welding current was 75 A, the strengthening phase was point-like or rod-like and the distribution was not uniform. When the welding current was increased to 80 A, due to the increase of welding heat input, the second phase had enough time to precipitate, but the shape was too coarse, which had an adverse effect on the quality of the weld. When the welding current was 85 A, the second phase shape rule was complete. When the welding current reached 90 A, due to excessive welding heat input, the precipitated strengthening phase remelted into the matrix, resulting in the deterioration of the mechanical properties of the welded joint.

When the welding current was small, the high-temperature residence time was short, the welding heat input was small and the aluminum alloy solidification effect was general while the weld was rapidly cooling, resulting in the second phase being too late for uniform precipitation, and the welded joint grain size was relatively small; with the increase in welding current, the size of the grains appeared to have a growth trend. This is mainly because the higher heat input increases the temperature of the molten pool of metal, allowing time for grain growth, and the grains grow rapidly.

### 4.5. Electrochemical Corrosion Properties

The polarization curves of the weld zone of the joint at different welding currents in a 3.5% NaCl solution under deoxygenation conditions are shown in Figure 16. From the figure, it can be understood that the polarization curve obtained when pitting repair is performed is divided into two types: one is a smooth transition curve, such as the curve at a welding current of 85 A. The other is the curve with inflection points, such as the welding current of the 75 A, 80 A, and 90 A curves. The Tafel extrapolation method was used to obtain the results of the fitting of the polarization curve as shown in Table 6. From the fitting results it can be seen that when the welding current was 90 A, the highest self-corrosion potential of the joint occurred. From the shape of the polarization curve, the specimens subjected to welding currents of 75 A, 80 A, and 90 A appeared as a passivation zone, whereas the specimens subjected to a welding current of 85 A did not show obvious signs of a passivation phenomenon occurring. This shows that when the welding current was 85 A, although the sample had a high corrosion potential, its passivation effect was not good. This is mainly related to the welding heat input, and the welding current was relatively small, resulting in a decreased over-age precipitation phase. Alloying elements in the matrix help to improve the corrosion potential, but the formation of passivation film may not be dense due to the influence of alloying elements, and thus no obvious passivation interval can be seen on the polarization cross-section.

When the corrosion potential in the dynamic potential polarization curve was more negative, the reaction equilibrium constant was smaller, and the resistance of the aluminum alloy in the electrochemical reaction process was smaller, leading to a higher likelihood of a corrosion reaction. When the welding current was 85 A, the corrosion potential at this time was the lowest, and thus this condition provided the best corrosion resistance. Overall, this is mainly related to the welding heat input. With different welding heat inputs, Mg_2_Si phase precipitation morphology, size and distribution are different. In the precipitation phase, due to the existence of potential differences between the collective, the process of corrosion will occur in the microcell reaction, thus changing the corrosion rate.

The main elements added to the 6063-T6 aluminum alloy are Mg and Si, and a small amount of Fe is also present. Because of the high content of elemental Si, the second phase in the specimen consists mainly of the precipitated phase, the Si phase and the Al(Fe)Si phase. Since only the corrosion potential of the precipitated phase is less than that of Al and the corrosion potential of the other second phase is greater than that of Al, two possibilities occur when the pitting of the material occurs. When the precipitated phase and the aluminum alloy matrix form a primary cell, the corrosion potential of the precipitated phase is lower, and pitting corrosion occurs in the precipitated phase. When the aluminum alloy matrix and another second phase with higher corrosion potential form a primary cell, pitting corrosion occurs in the aluminum alloy matrix.

The corrosion current and corrosion potential in the dynamic potential polarization curve reflects the kinetic parameters of corrosion rate and the thermodynamic parameters of material corrosion tendency, respectively. A lower corrosion current density indicates that the material is more resistant to corrosion and a lower corrosion potential indicates that the material is more susceptible to corrosion. After MIG welding of 6063-T6 aluminum alloy, the microstructure of the weld area of the joint changes significantly, which leads to a change in its corrosion resistance.

When the welding current is small, small holes appear on the surface of the weld area of the joint, and these holes are more susceptible to the erosion of Cl^−^ in the NaCl solution, resulting in a decrease in the corrosion resistance of the welded joint. It has been shown that the dissolution of Al, Mg and Si elements in aluminum alloys is due to the difference in pitting potential [34].

As shown in Figure 17, different degrees of pitting occurred on the surface of welded joints. The pitting degree gradually deepened and spread to all sides with the increase of welding current, and the pits also gradually grew larger. Compared with the other three groups of parameters, the joint corrosion at 85 A welding current was not serious: only a small number of pits appeared, and the joint had good corrosion resistance, which is also echoed in the previous research results.

Figure 18a,c show the morphological characteristics of the pits and their surroundings. There are white corrosion residues of different shapes in the pits, and the black precipitation phase is predominant near the pits and corrosion cracks also appear around them, so the presence of the precipitation phase in the welded joint will affect the corrosion of the joint. Figure 18b,d show the results of EDS analysis, through which it can be found that the black precipitates are more abundant in Mg and Si elements, and the white precipitates are more abundant in Fe elements. The pitting behavior of welded joints essentially occurs on the surface of Mg_2_Si particles, and the black precipitation phase Mg_2_Si dominates in the pits. With the elevation of welding current, the pitting corrosion gradually increases, and the self-corrosion potential of Mg_2_Si is lower than that of other phases, and thus it will be the first to corrode, and the presence of Mg_2_Si phase will seriously deteriorate the corrosion resistance of welded joints. The Si phase will remain in the crater, and the self-corrosion potential of the Al substrate is higher than that of the Si phase, so it will again accelerate the corrosion of the surrounding Al substrate [35].

## 5. Conclusions

This article uses finite element technology to successfully simulate the 6063-T6 aluminum alloy MIG welding process, through tensile experiments, electrochemical corrosion and other means to study the impact of welding current on the mechanical properties of welded joints and microstructural changes in the law. This study reaches the following conclusions:The macroscopic morphology of the weld front reaches its optimum when the welding current is 85 A.The combined heat source model of double ellipsoid heat source and Gaussian body heat source is more suitable for the numerical simulation of 6063-T6 aluminum alloy MIG welding.When the welding current increases from 75 A to 85 A, the precipitation phase gradually becomes larger; when the welding current reaches 90 A, the precipitation phase from the new solid solution to the internal body occurs and the grain continues to grow, resulting in a decrease in the mechanical properties of the joint.With the increase in welding, the dimple size of the tensile fracture tends to increase, and the depth becomes shallow, which leads to the weakening of mechanical properties. The fracture morphology is consistent with ductile fracture characteristics. When the welding current is 85 A, the maximum tensile strength is 110.9 MPa and the elongation is 16.3%, the Vickers Microhardness reaches the highest value of 46.9 HV.The pitting degree of the joint weld zone is serious with the increase of welding current. When the welding current is 85 A, E_corr_ = −0.678 V, I_corr_ = 2.259 × 10^−8^A/cm^2^, the corrosion resistance of the welded joint is optimum. When the welding current is 90 A, E_corr_ = −1.124 V, I_corr_ = 4.133 × 10^−8^ A/cm^2^, the corrosion resistance is the worst.

## Figures and Tables

**Figure 1 materials-16-04886-f001:**
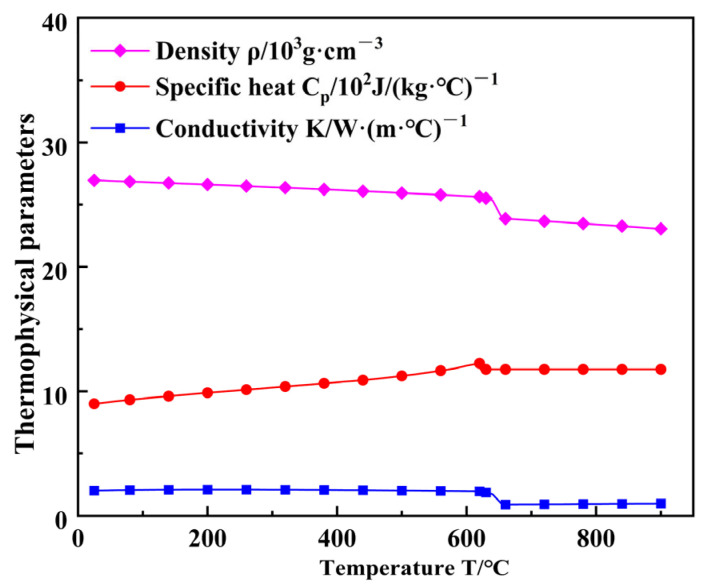
Thermal conductivity, specific heat capacity and density of 6063-T6 aluminum alloy.

**Figure 2 materials-16-04886-f002:**
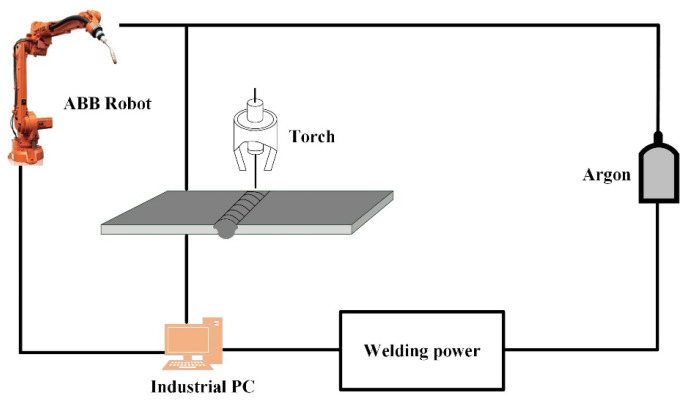
Schematic diagram of MIG welding.

**Figure 3 materials-16-04886-f003:**
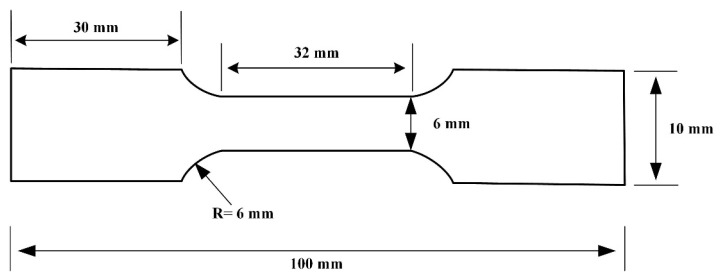
Tensile Specimen.

**Figure 4 materials-16-04886-f004:**
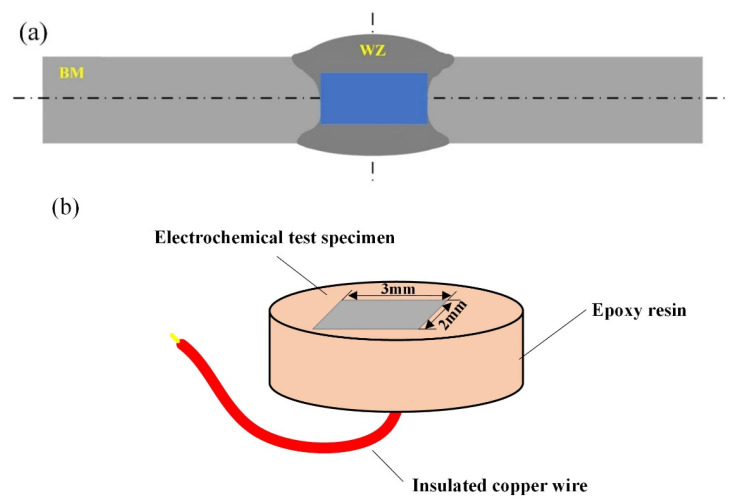
Schematic diagram of electrochemical specimens. (**a**) Sampling position; (**b**) electrochemical specimen.

**Figure 5 materials-16-04886-f005:**
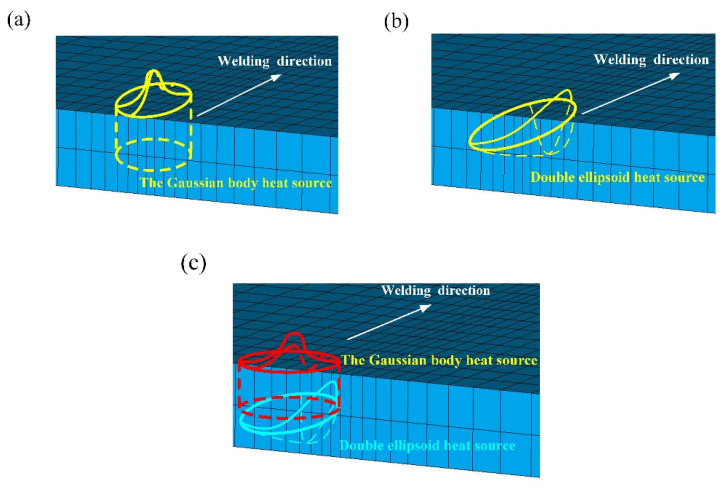
Schematic diagram of heat source loading: (**a**) Gaussian body heat source, (**b**) double ellipsoid heat source, (**c**) double ellipsoid heat source + Gaussian body heat source.

**Figure 6 materials-16-04886-f006:**
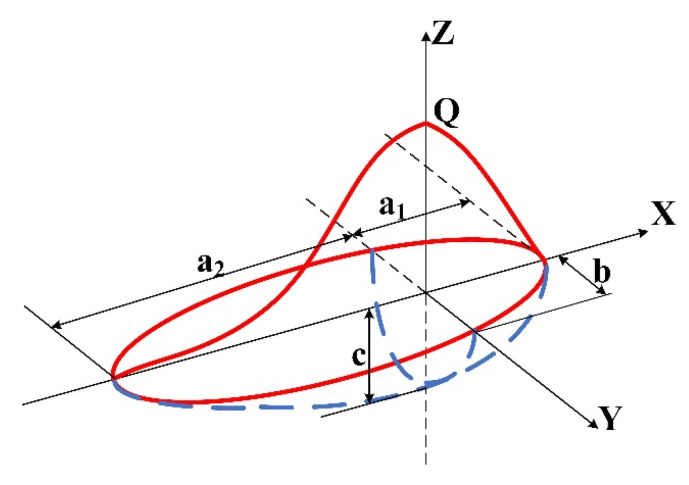
Schematic diagram of the double ellipsoid heat source model.

**Figure 7 materials-16-04886-f007:**
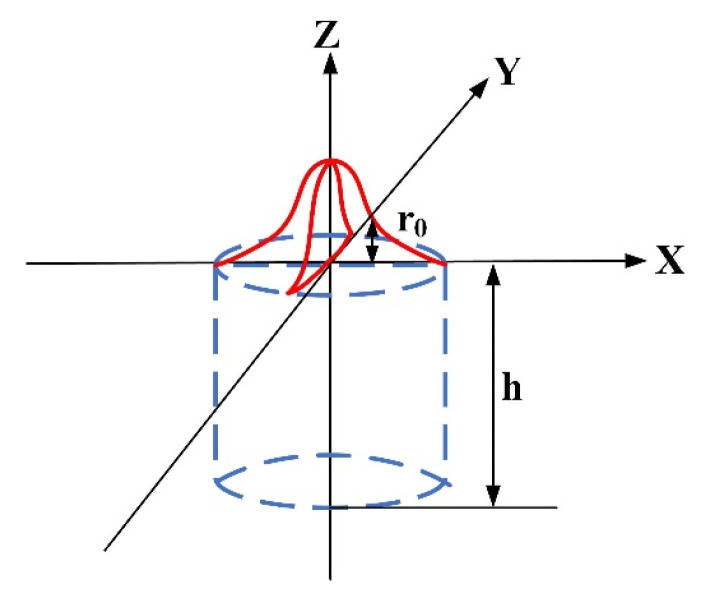
Gaussian body heat source model.

**Figure 8 materials-16-04886-f008:**
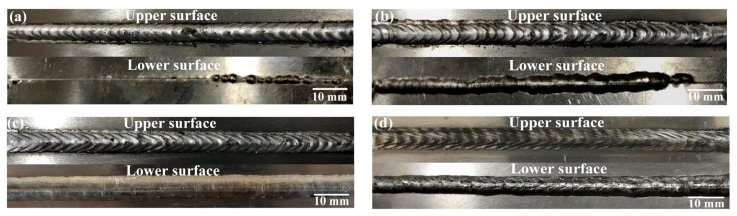
Macroscopic morphologies of MIG welds at different welding currents: (**a**) 75 A, (**b**) 80 A, (**c**) 85 A, (**d**) 90 A.

**Figure 9 materials-16-04886-f009:**
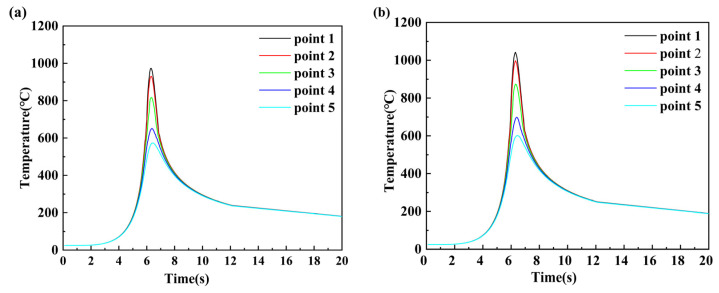
Thermal cycle curves at various points at different welding currents: (**a**) 75 A, (**b**) 80 A, (**c**) 85 A, (**d**) 90 A.

**Figure 10 materials-16-04886-f010:**
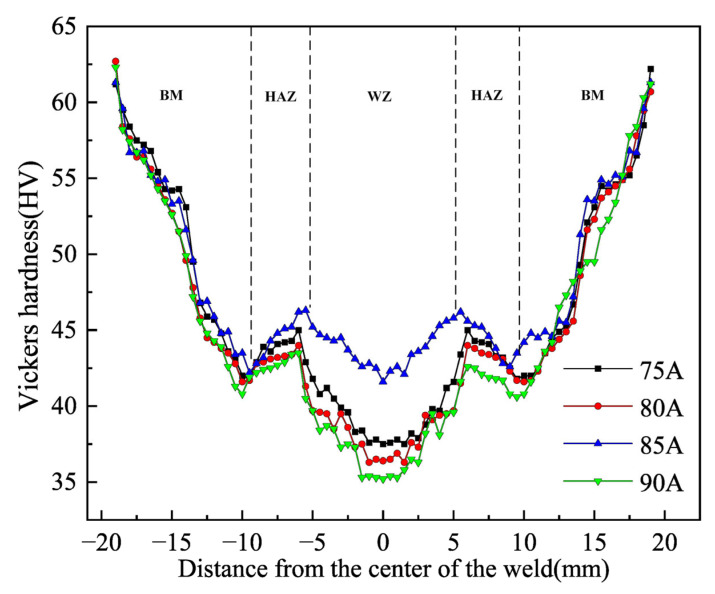
Microhardness of welded joints under different welding currents.

**Figure 11 materials-16-04886-f011:**
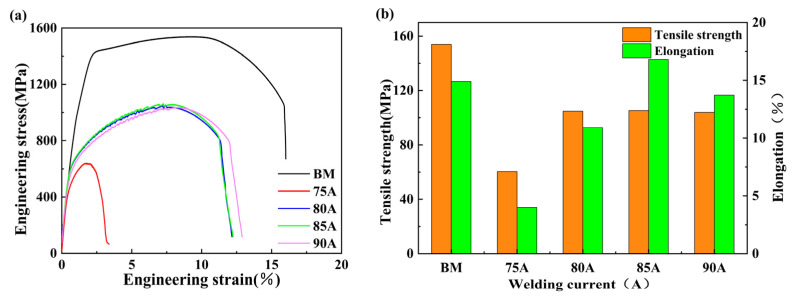
Tensile properties of base metal and specimens at different welding currents: (**a**) tensile curve, (**b**) tensile test results, (**c**) tensile samples used for the experiment, (**d**) the fracture position of the tensile specimen.

**Figure 12 materials-16-04886-f012:**
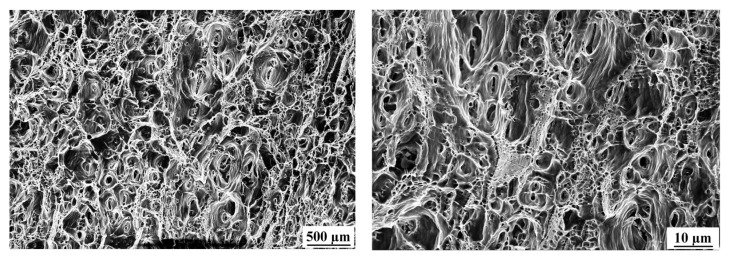
SEM scan image of tensile fracture of base material.

**Figure 13 materials-16-04886-f013:**
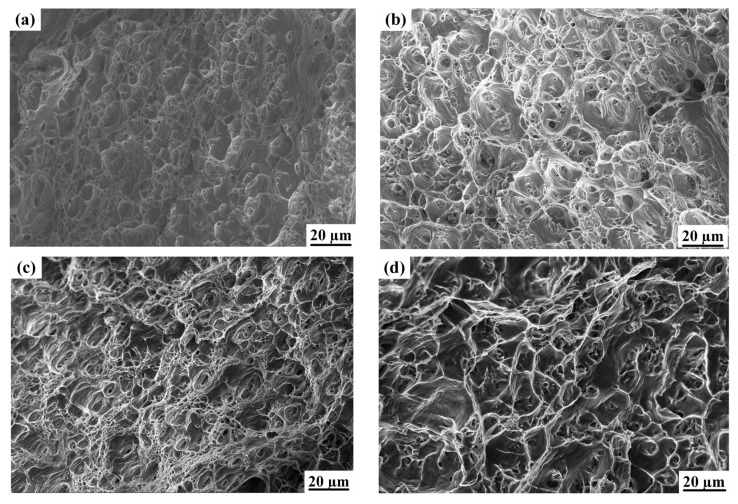
The shape of the fracture at different welding currents: (**a**) 75 A, (**b**) 80 A, (**c**) 85 A, (**d**) 90 A.

**Figure 14 materials-16-04886-f014:**
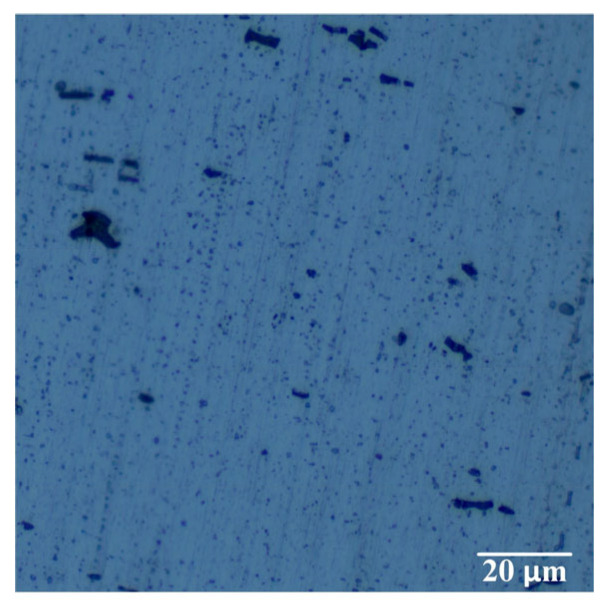
Microstructure of 6063-T6 aluminum alloy base metal.

**Figure 15 materials-16-04886-f015:**
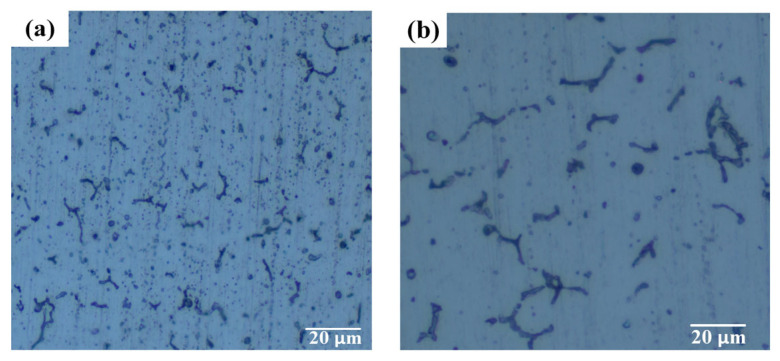
Microstructure of 6063-T6 aluminum alloy weld: (**a**) 75 A, (**b**) 80 A, (**c**) 85 A, (**d**) 90 A.

**Figure 16 materials-16-04886-f016:**
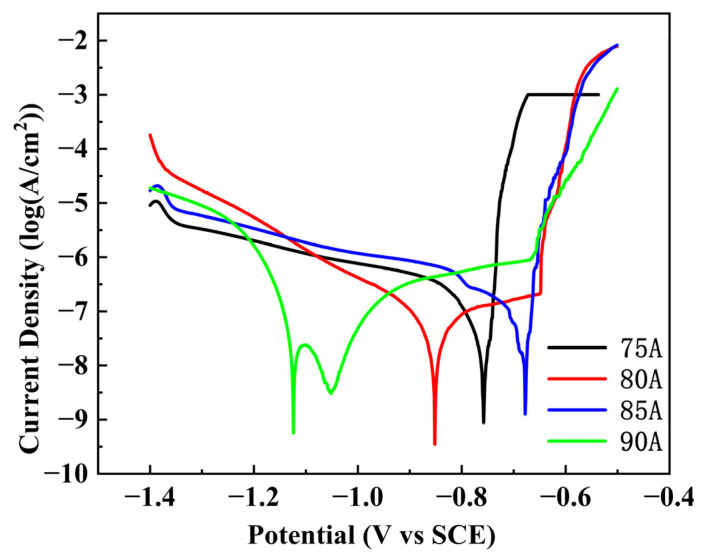
Polarization curves in 3.5% NaCl solution in the WZ under different welding currents.

**Figure 17 materials-16-04886-f017:**
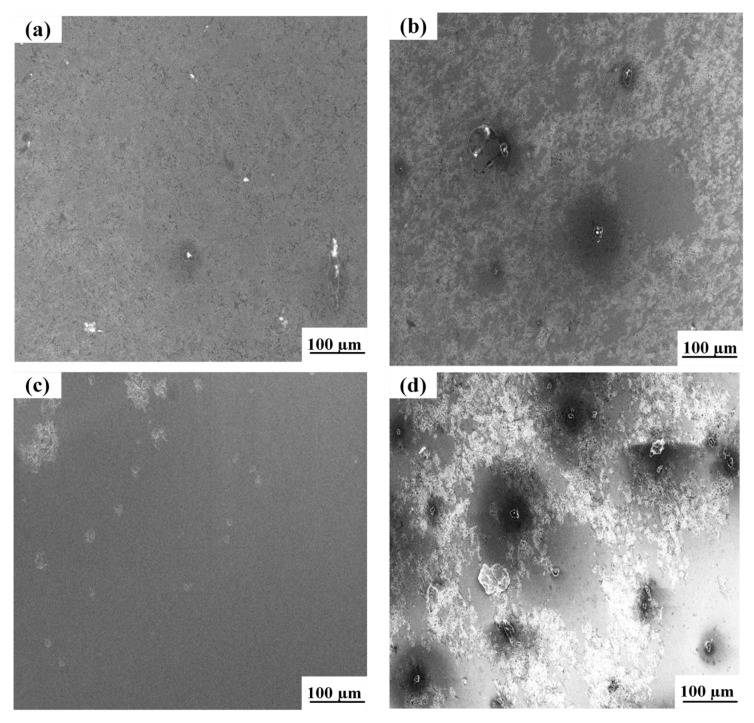
Microstructure of the weld zone after pitting at different welding currents: (**a**) 75 A, (**b**) 80 A, (**c**) 85 A, (**d**) 90 A.

**Figure 18 materials-16-04886-f018:**
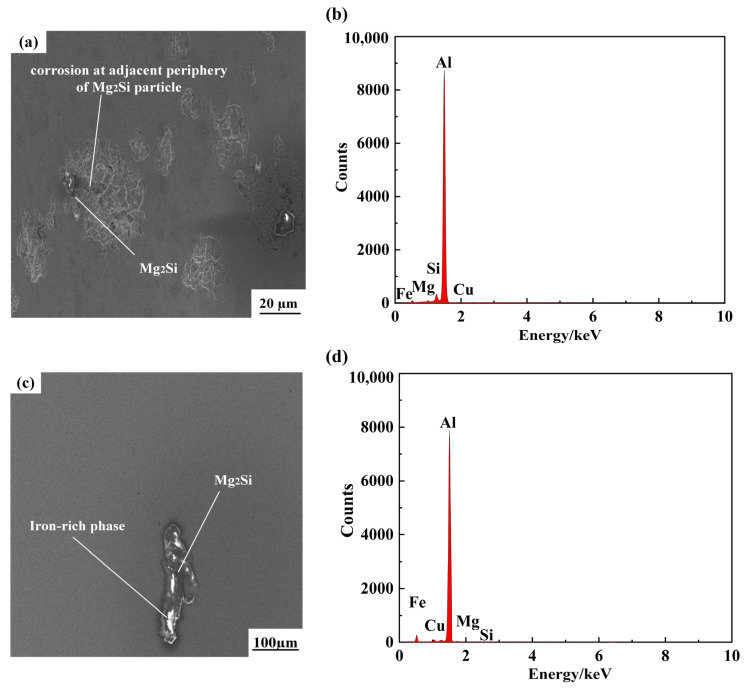
SEM morphology of pits and EDS analysis results. (**a**) Black precipitation phase, (**b**) EDS pattern of black precipitation phase, (**c**) white precipitation phase, (**d**) EDS pattern of white precipitation phase.

**Table 1 materials-16-04886-t001:** Chemical composition of 6063-T6 aluminum alloy and ER5356 welding wire (wt.%).

Material	Mn	Mg	Si	Cu	Cr	Ti	Zn	Fe	Al
6063-T6	0.009	0.707	0.381	0.004	0.002	0.029	0.003	0.244	Re.
ER5356	0.15	5	0.04	0.01	0.1	0.1	0.01	0.1	Re.

**Table 2 materials-16-04886-t002:** Mechanical properties of 6063-T6 aluminum alloy and ER5356 wire.

Material	Yield Strength (MPa)	Tensile Strength (Mpa)	Elongation (%)
6063-T6	216	253	15
ER5356	258	290	21

**Table 3 materials-16-04886-t003:** Welding Parameters.

Test Number	Welding Current (A)	Gas Flow (L/min)	Welding Efficiency	Welding Speed (cm/min)
1	75	20	0.75	50
2	80	20	0.75	50
3	85	20	0.75	50
4	90	20	0.75	50

**Table 4 materials-16-04886-t004:** Numerical simulation of welding pool shape and the actual shape of the comparison.

Name	Welding Pool Shape Comparison	Temperature
Double ellipsoid heat source	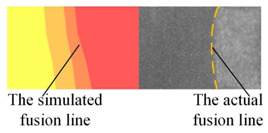	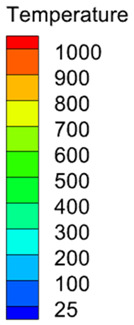
Gaussian body heat source	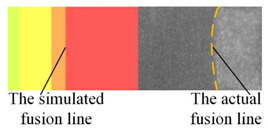	
Gaussian body heat source + double ellipsoid heat source	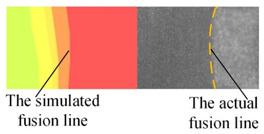	

**Table 5 materials-16-04886-t005:** Tensile test results of tensile samples under different welding currents.

Welding Parameters	Tensile Strength (MPa)	Elongation (%)	Joint Efficiency (%)
75 A	50	3	32
80 A	106	11	68
85 A	108	17	69
90 A	105	13	66

**Table 6 materials-16-04886-t006:** WZ of the joint at different welding currents in 1.5 mol/L NaCl solution polarization curve fitting results.

Welding Current/A	E_corr_/V	I_corr_/Log (A/cm^2^)
75	−0.756	3.775 × 10^8^
80	−0.852	9.280 × 10^8^
85	−0.678	2.259 × 10^8^
90	−1.124	4.133 × 10^8^

## Data Availability

Not applicable.

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
