# Peer review of "Study on Morphology, Microstructure and Properties of 6063-T6 Aluminum Alloy Joints in MIG Welding"

_materials, 2023, doi:10.3390/ma16134886_

Round 1

Reviewer 1 Report

I suggest that the authors must  take into account the following corrections/suggestions:

1. The article is well-written, but there are some grammatical and typo errors; the paper should be free from all types of errors.

2.  Polish the abstract by presenting some significant and key outcomes.

3. The introduction section should have some most recent and new publications.

4. Revise the results and discussion section briefly; I think you are missing some results here. Also, give some detail about the figures.

5. Include more outcomes of your study in the conclusions and future directions.

It requires to improve the language. Some sentences are not complete, and somes are missed.

Author Response

We appreciate your comments and have made some improvements to the original response. Please see the attachment.

Reviewer 2 Report

Review report: Study on morphology, microstructure and properties of 6063-T6 aluminum alloy joints in MIG welding: 1. Abstract: Add some quantitative results related to mechanical testing at the end of the abstract section. Shorten the length of the conclusion section. 2. Introduction: In place of citing multiple references, explain the individual work of the author and try to make a bridge between current and previous work. 3. Novelty and application: Add a separate section for novelty and application of work. 4. Please check the values in Table 1 and Table 2. 5. Also add reference or detail on how they were obtained. 6. Add reference for Fig. 1. 7. Add the image of each specimen prepared for the mechanical testing and also add the standard and machine used for their testing. 8. Add an image of the actual setup. 9. Add a reference for each equation. 10. Add complete detail of the model and boundary conditions. 11. Scale required in a few images. 12. Add an image of the fractured tensile specimen, and fracture location and joint efficiency in separate tables: https://doi.org/10.1016/j.jmapro.2019.10.002 13. Please provide clear discussion about cleavage and dimples and also discuss the mechanism related to the failure of the specimen by including the macrograph of the fractured tip: https://doi.org/10.1016/j.engfailanal.2018.09.036. 14. Mention the region of weld and HAZ in the hardness plot. 15. Try to relate the microstructure and mechanical properties. 16. Add technical discussion regarding the corrosion test.

NA

Author Response

(The authors gave the same response as above.)

Reviewer 3 Report

Authors should address and correct the following issues in the paper:

Abstract - what does the term "perfect welding" refer to? There is no such thing as a perfect weld, perhaps the authors meant to say "high-quality welding"?

There is also no mention of TIG welding, which is commonly used for aluminium alloys. Some comparison between this procedure and MIG could be added to the introduction

Introduction

The sentence which lists the applications of the aluminium alloy in question could use some more references, although this is not mandatory to include

When talking about advantages of MIG welding, authors could add that it is also very productive and efficient

Experimental procedure

filler material ER5356 is named according to which standard? This could also be included as a reference

Also, chemical composition of the base material and filler material are a bit different. There should be a more detailed explanation how the filler material was selected

Table 2

Something about these numbers is off... Did the authors accidentally copy the values from Table 1, because 0.15 and 5 correspond to Mn and Mg contents...

Electrochemical experiments

The type of corrosion that was considered in this study is not mentioned in the introduction, just add that it was electrochemical

Figure 8

According to the image in fig. 8c, there seems to be some porosity in the upper surface... if not, what are the black spots which are clearly visible in some parts of the weld face?

4.2. Numerical simulations...

Chapter 4.2.2 is a bit confusing, are you still talking about the numerical results? Or are the measuring points for temperature obtained experimentally?

Either way, a detailed comparison between simulations and the experiments seems to be missing, beyond the point where it was concluded that 85A current is the most ideal one.

This, or the previous section, should contain some more details about the numerical simulations

Table 5

What does WZ in this table stand for?

In general, there is no mention of the heat affected zone of the welded joints, despite a well-explained, detailed microstructural analysis presented in the paper. Since the heat affected zone represents a very important welded joint region, due to it often being its weakest part, it is common practice to consider it separately from the rest of the weld.

The authors could include more detailed information about the HAZ, or alternatively, explain why they did not take it into account (since the analysis presented looks pretty good even without it...)

Some minor grammar changes should be made to the text as a whole, mostly in the introduction and some of the final sections (like the corrosion chapter...)

Author Response

(The authors gave the same response as above.)

Reviewer 4 Report

Manuscript is good and can be accepted after the following improvement:

- improve the abstract.

-add relevant literature, this literature could be helpful "Quality Assessment and Features of Microdrilled Holes in Aluminum Alloy Using Ultrafast Laser. In Light Metals 2023 (pp. 380-386). Cham: Springer Nature Switzerland"; "Mechanical and microstructural investigation of dissimilar joints of Al-Cu and Cu-Al metals using nanosecond laser. Journal of Mechanical Science and Technology36(8), pp.4205-4211".

-What is the novelness of the research work and how does it affect the scientific comunity.

Please check the typo error.

Author Response

(The authors gave the same response as above.)

Reviewer 5 Report

I would like to congratulate the authors for the thorough research and the article, especially in the field of pitting corrosion on the weld bead made with various welding currents. As such, without any doubt, I am in favor of publishing the article but with some minor corrections:

1. In line 145 the term "t is time" appears. Unfortunately, the term t is not involved in the corresponding mathematical relationship (line 143).

2. In lines 154 and 153 the term "hc" appears. Unfortunately, the term hc is not involved in the corresponding mathematical relationship (line 151).

3. On line 188, in the explanations given, instead of the phrase "...and x2+y2 is the distance..." I would put "...and x2+y2 is the distance...",

4. I believe that in point 4.2.1 (starting with line 209) it should be explained in more detail the place from where the temperature field is visualized in "Welding pool shape comparison" of table 4 because I did not understand. It doesn't mean it's not useful and valuable, it just needs to be explained in more detail.

5. In table 4 (line 218) a yellow dot line appears (in column ”Welding pool shape comparison”), interrupted between two visually distinct domains. What is it trying to suggest?

6. Starting with line 310 (Fig 14) make a thorough explanation of the phases involved in the microstructure. In my opinion, this is a key-point of the article, which needs to be strongly argued. I don't think the black particles are the Mg2Si phase; an EDS analysis should performed to prove the claim,

7. In general, all the literature of lines 311-334 contains only some assumptions. Their demonstration would have required EDS analyzes (similar to those in Fig. 18)

8. On line 397 instead of "Figures 18a and 18c shows..." should be put "Figures 18a and 18c show..."

In general the English used was good

Author Response

(The authors gave the same response as above.)
